# Trends in Gastrointestinal Infections during the COVID-19 Pandemic and Concerns of Post-Pandemic Resurgence in Japan

**DOI:** 10.3390/diseases12010004

**Published:** 2023-12-21

**Authors:** Takuma Higurashi, Shigeki Tamura, Noboru Misawa, Nobuyuki Horita

**Affiliations:** 1Department of Gastroenterology and Hepatology, Yokohama City University School of Medicine, Yokohama 236-0004, Japan; tamura.shi.gs@yokohama-cu.ac.jp (S.T.); nobomisa@yokohama-cu.ac.jp (N.M.); 2Chemotherapy Center, Yokohama City University Hospital, Yokohama 236-0004, Japan; horitano@yokohama-cu.ac.jp

**Keywords:** gastrointestinal infections, COVID-19, epidemiology, trends

## Abstract

The impact of the COVID-19 pandemic was very broad and substantial, affecting a variety of fields worldwide. In Japan, the infection began spreading in March 2020. At that time, the government alerted people to cancel overseas travel, and encouraged wearing of masks, handwashing, sanitizing and keeping social distance. We sought to determine how COVID-19 infections affected other infectious diseases by investigating the trends in seven gastrointestinal infections that are listed among the 77 important infectious diseases designated by the National Institute of Infectious Diseases. We compared seven gastrointestinal infectious diseases, namely cholera, bacterial dysentery, enterohemorrhagic Escherichia coli, typhoid fever, paratyphoid fever, amoebic dysentery, and giardiasis, in terms of numbers of new cases before the COVID-19 pandemic (2012–2019) and during the pandemic (2020–2022). During the COVID-19 pandemic period (2020–2022), the incidence of the seven infections decreased significantly (*p* < 0.05) compared with before the pandemic (2012–2019). The sharp and significant decline in incidence of these seven infections in Japan during the COVID-19 pandemic period (2020–2022) appears to be due to restrictions on overseas travel and strict anti-infection measures, such as self-quarantine and encouragement of handwashing and sanitizing. The number of new cases of gastrointestinal infections in Japan is expected to increase in 2024 as these measures lapse. It is important for physicians to continue to monitor trends in gastrointestinal infections and educate people about proper infection prevention.

## 1. Introduction

The emergence of COVID-19 in December 2019 marked the beginning of a global health crisis, with the disease swiftly spreading worldwide, ultimately culminating in a pandemic [1,2]. Japan, like many other nations, faced the challenges of this unprecedented situation. In March 2020, the country began grappling with the increasing spread of the virus, prompting the government to declare a state of emergency in April 2020. This critical response was accompanied by a plea for self-quarantine, as COVID-19’s characteristics were still largely unknown, and it posed a significant threat with its high fatality rate. In the absence of vaccines, health authorities in Japan swiftly implemented a range of preventive measures, including the promotion of mask-wearing, frequent handwashing, sanitizing practices, and the imperative need for social distancing. These measures were crucial in curbing the spread of the virus. However, these proactive measures also had various unintended consequences, creating a multifaceted impact on the healthcare system. During this period, there was a notable decrease in hospital visits and medical examinations, a phenomenon observed across the healthcare spectrum [3,4,5,6,7,8,9,10]. This decline had repercussions on various aspects of healthcare, particularly in the realm of chronic diseases. For instance, the number of newly diagnosed heart failure and stroke patients declined, while the proportion of severely ill patients increased. These shifts in patient demographics and disease severity could be attributed to decreased numbers of consultations for patients with relatively minor illnesses and symptoms [11,12,13].

Conversely, as people delayed hospital visits due to pandemic-related fears or lifestyle changes, there was an unexpected uptick in newly diagnosed diabetes patients. Similarly, diabetic ketoacidosis cases surged, suggesting that delayed medical attention, altered lifestyles, and heightened stress levels contributed to these trends [13]. Another startling revelation was the decline in newly diagnosed cancer cases during the lockdown. Reports emerged highlighting this concerning trend, raising questions about the impact of reduced medical consultations and screening programs on cancer detection rates [14].

In light of these profound changes and challenges posed by the COVID-19 pandemic on healthcare, we sought to unravel the broader implications by investigating how the pandemic influenced the rates of other infectious diseases in Japan. Our focus was on seven gastrointestinal infections, categorized among the 77 important infectious diseases designated by the National Institute of Infectious Diseases in Tokyo, Japan. This comprehensive study aims to shed light on the far-reaching health consequences and valuable lessons to be learned from the ripple effects of the COVID-19 pandemic across various aspects of healthcare.

## 2. Materials and Methods

### 2.1. Study Design and Setting

This was a retrospective database analysis study to investigate the trends in important gastrointestinal infectious diseases that are designated as important infectious diseases by the National Institute of Infectious Diseases. The requirement for patient consent was waived due to all data already being anonymized. This report followed the Strengthening the Reporting of Observational Studies in Epidemiology (STROBE) guidelines. The study complied with the Declaration of Helsinki [15] and the Ethics Guidelines for Clinical Research published by the Ministry of Health, Labor, and Welfare, Japan [16].

### 2.2. Patients and Data Extraction: Primary Outcome

In Japan, the Infectious Diseases Law instructs physicians to report all cases of 77 designated important infectious diseases. These 77 diseases include seven gastrointestinal infections with mandatory reporting, namely cholera, bacterial dysentery, enterohemorrhagic Escherichia coli, typhoid fever, paratyphoid fever, amoebic dysentery, and giardiasis. The National Institute of Infectious Diseases provided us with this incidence data for those seven diseases [17]. We compared the numbers of new cases reported in the period before the COVID-19 pandemic (2012–2019) versus those during the pandemic (2020–2022).

### 2.3. Statistical Analysis

Statistical analyses were performed using the Mann–Whitney U test. *p*-values of <0.05 were regarded as denoting statistical significance. The analysis was performed using SPSS software, version 27.0 (SPSS, Chicago, IL, USA).

## 3. Results

Numbers of Gastroenteritis Cases before and during the COVID-19 Pandemic.

In the case of cholera, the data revealed a stark contrast in the median annual number of newly diagnosed patients in Japan before the onset of the COVID-19 pandemic, which stood at 5 (with a range of 3–9 cases), and during the pandemic, when this number dropped to a median of 1 (with a range of 0–1 cases).

For bacterial dysentery, the situation mirrored this trend, with a median of 149.5 newly diagnosed patients annually before the COVID-19 era, ranging from 140 to 268 cases. However, during the pandemic, the median plunged to 16 cases, with a range of 7–87, indicating a substantial decrease in cases.

Similarly, for enterohemorrhagic *Escherichia coli*, the median annual number of newly diagnosed patients in Japan prior to COVID-19 was 3811 (with a range of 3573–4151 cases). However, during the pandemic, this number decreased to a median of 3243 (with a range of 3094–3352 cases), reflecting a notable reduction.

The trend continued with typhoid fever, which exhibited a median annual number of 37 newly diagnosed patients before COVID-19, ranging from 35 to 65 cases. During the pandemic, this median dropped to 17 cases, with a range of 4–21 cases.

Paratyphoid fever also followed a similar pattern, with a median annual number of 22 newly diagnosed patients before COVID-19 (ranging from 14 to 50 cases) and a median of 7 cases during the pandemic, with a range of 0–9 cases.

Amoebic dysentery experienced a substantial decrease in incidence during the pandemic. The median annual number of newly diagnosed patients before COVID-19 was 1086 (with a range of 843–1151 cases), whereas during the pandemic, the median dropped to 537 cases, with a range of 529–611 cases.

Lastly, giardiasis exhibited a similar pattern, with a median annual number of 69.5 newly diagnosed patients before COVID-19 (ranging from 53 to 81 cases). During the pandemic, the median decreased to 30 cases, with a range of 28–32 cases.

These findings, as shown in Figure 1 [17], clearly highlight that each of these infectious diseases experienced a significant and consistent decrease in incidence during the COVID-19 pandemic when compared to the pre-pandemic period (*p* < 0.05 for each).

## 4. Discussion

The significant decrease in the incidence of the seven gastrointestinal infections among the 77 designated important infectious diseases during the COVID-19 pandemic era (2020–2022) in Japan compared to the pre-pandemic period (2012–2019) underscores the profound impact of the pandemic on public health. These substantial declines can be attributed, at least in part, to the rigorous measures implemented to combat the spread of COVID-19. These measures included stringent travel restrictions, the enforcement of comprehensive anti-infection protocols such as self-quarantine, and the widespread promotion of meticulous handwashing practices. The effectiveness of these measures in reducing the transmission of not only COVID-19 but also other infectious diseases cannot be underestimated.

As the fight against COVID-19 progressed, it marked a series of significant milestones in Japan. Both the World Health Organization and the Japanese government made the decision to lift the “State of Emergency” declaration for COVID-19 in May 2023. This was accompanied by a reclassification of COVID-19 under the Infectious Diseases Control Law from Category II, which required strict infectious control measures, to Category V, indicating that strict infectious control measures were no longer deemed necessary within the country. These transitions were made possible by the development and distribution of several effective vaccines, the evolving nature of viral strains, and a declining mortality rate. These factors collectively paved the way for a gradual return to normalcy, marked by increased social activities and the resumption of overseas travel [18].

Such declines have also been observed in other countries as well. For example, in England, compared with the 5-year average (2015–2019), during the first 6 months of the COVID-19 response, there was a 52% decrease in gastrointestinal disease outbreaks reported and a 34% decrease in laboratory-confirmed cases [19]. These trends have also been observed in Korea and the United States [20]. They concluded that if some of these changes in behavior such as improved hand hygiene were maintained, then we could potentially see sustained reductions in the burden of gastrointestinal illness.

However, it is important to note that during the height of the pandemic, there was a significant decrease in the number of visitors to Japan, a trend that correlated with the observed reduction in gastrointestinal infections. Nevertheless, as people have become more eager to travel between regions and have started to relax their adherence to stringent handwashing practices, the number of visitors to Japan is once again on the rise. Regrettably, this resurgence in travel activity is expected to be accompanied by an increase in the incidence of new cases of gastrointestinal infections in Japan in 2024. This underscores the continued importance of healthcare professionals remaining vigilant and closely monitoring trends in gastrointestinal infections. Additionally, it emphasizes the ongoing need for public education on proper infection prevention measures to mitigate the risk of disease transmission.

On the other hand, for diseases other than infectious diseases, such as cancer [21], the number of diagnoses also declined markedly during the COVID-19 pandemic, but has recovered since the end of the pandemic due to the resumption of medical services and changes in patient visitation behavior. It is therefore important to focus on the study of the impact of COVID-19 not only on infectious diseases but also on other diseases, because these findings may become useful in predicting future pandemics. While this study has provided valuable insights, it is crucial to acknowledge its limitations. First, the dataset used for analysis lacks information about the age or sex of the subjects, or the severity of the diseases; it solely presents the number of affected patients. Secondly, the analysis focused primarily on comparing the numbers of patients and did not account for potential confounding factors beyond the influence of COVID-19. Nonetheless, it is worth highlighting that these gastrointestinal infections often exhibit characteristics associated with imported infections. Furthermore, the fact that other designated important infectious diseases, such as viral hepatitis, acquired immunodeficiency syndrome (AIDS), measles, and rubella, did not exhibit the same declining trend underscores the strong likelihood that the rapid decrease in these gastrointestinal infections is indeed closely linked to the combined impact of travel restrictions and the stringent infection control measures implemented in response to COVID-19. 

## 5. Conclusions

The sharp and significant decline in incidence of these seven gastrointestinal infections in Japan during the COVID-19 pandemic period (2020–2022) appears to be due to restrictions on overseas travel and strict anti-infection measures, such as self-quarantine and encouragement of handwashing and sanitizing. It will be important for physicians to continue to monitor trends in gastrointestinal infections and educate people about proper infection prevention.

## Figures and Tables

**Figure 1 diseases-12-00004-f001:**
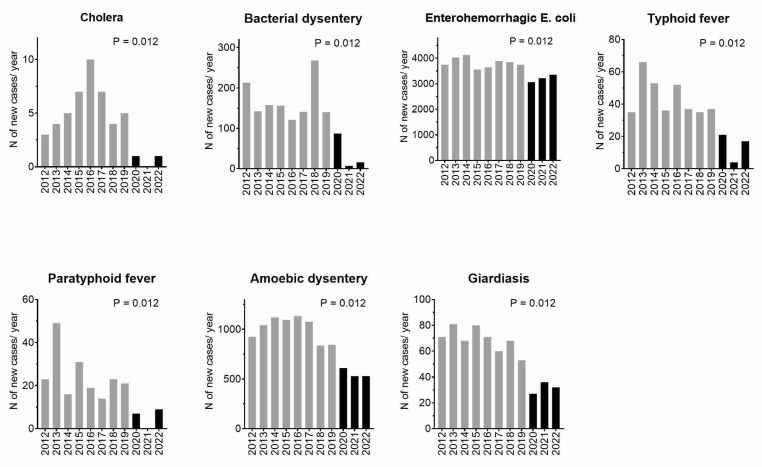
Numbers of gastroenteritis cases before and during the COVID-19 pandemic: the gray bars represent the pre-pandemic era from 2012 to 2019, while the black bars represent the pandemic era from 2020 to 2022. *p*-values based on Mann–Whitney U tests were used to compare the pre-pandemic and pandemic eras.

## Data Availability

The datasets used and analyzed during the study are available from the corresponding author upon reasonable request and are available from the National Institute of Infectious Diseases website. https://www.niid.go.jp/niid/ja/data.html (accessed on 1 November 2023).

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
