# Peer review of "Trends in Gastrointestinal Infections during the COVID-19 Pandemic and Concerns of Post-Pandemic Resurgence in Japan"

_diseases, 2023, doi:10.3390/diseases12010004_

Round 1

Reviewer 1 Report

Comments and Suggestions for Authors

In the presented article, only the number of diseases is compared. Figures should be standardized according to age and gender, and significance should be given by comparing the standardized morbidity ratio. The statement that these diseases will increase in the coming years is extremely subjective. Its validity is controversial. Disease Rates should be projected using the estimated regression analysis method, giving the forecast for the years 2024 and 2027.

The MW U test is extremely inadequate in terms of analysis of disease numbers.

​

Author Response

We appreciate your insightful comments on our statistical methods.

As you correctly point out, the Mann–Whitney U test is not the optimal method for predicting future trends in the number of patients with a specific disease. However, the main objective of our manuscript is not to estimate the cases of future gastrointestinal infections; rather, we aimed to compare the numbers between the pre-pandemic and during-pandemic eras. We have altered the language to avoid giving the impression that this was a predictive exercise.

We acknowledge your suggestion that linear regression analysis is valuable for forecasting future trends based on available data. Although gastrointestinal infections decreased during the COVID-19 period, we anticipate an increase as the pandemic subsides, given the increased frequency of communication and travel without adherence to infection prevention measures. Considering this potential for a reversal in trends, unfortunately, applying linear regression may not be advisable.

We have added these viewpoints in the Discussion, notably in the last paragraph before the Conclusions. Once again, we sincerely appreciate your valuable feedback.

Reviewer 2 Report

Comments and Suggestions for Authors

The authors compare the frequency of significant infections during the COVID-19 pandemic and the post-pandemic period. However, the results were as anticipated due to the increased movement of people and goods, along with the relaxation of behavioral restrictions.

To broaden the discussion, I believe additional information is required:

1: What is the frequency and tendency of infectious diseases in comparison to other countries?

2: What is the likelihood of the future spread of other infectious diseases?

I contend that addressing these questions would provide valuable insights and enhance the overall understanding of the topic.

Author Response

We thank the reviewer for these useful comments. Per your suggestion, we have added references and discussion of some additional studies of the impact of the COVID-19 pandemic on gastrointestinal infections in several countries (References 19-21). These reports on trends of gastrointestinal infections during the COVID-19 pandemic in other countries were similar to and supportive of our findings in Japan. We have added these viewpoints in the Discussion. We also added other viewpoints about COVID-19 and newly diagnosed cancers.

              Regarding the likelihood of the future spread of other infectious diseases, our dataset is unfortunately inadequate to provide useful predictive information on this topic. We have addressed this limitation in the last paragraph of the Discussion.

With these additions, we believe that this revised manuscript is much improved and hope that it is now acceptable for publication.

Round 2

Reviewer 1 Report

Comments and Suggestions for Authors

What is important here is not the number of diseases, but the rates specific to the number of diseases. If it is thought that population movements in Japan do not change over the years, this form can be accepted.

Reviewer 2 Report

Comments and Suggestions for Authors

The authors believe that the points raised have been adequately addressed. No further revisions are requested.